# High-Frequency Gaseous and Particulate Chemical Characterization using Extractive Electrospray Ionization Mass Spectrometry (Dual-Phase-EESI-TOF)

Chuan Ping Lee[1], Mihnea Surdu[1], David M. Bell[1], Josef Dommen[1], Mao Xiao[1], Xueqin Zhou[1], Andrea Baccarini[1,2], Stamatios Giannoukos[1,3], Günther Wehrle[1], Pascal André Schneider[1], Andre S. H. Prevot[1], Jay G. Slowik[1,*], Houssni Lamkaddam[1,*], Dongyu Wang[1,*], Urs Baltensperger[1], Imad El Haddad[1,*]

[1]Laboratory of Atmospheric Chemistry, Paul Scherrer Institute (PSI), 5232 Villigen, Switzerland
[2]School of Architecture, Civil and Environmental Engineering, École Polytechnique Fédérale de Lausanne, Switzerland
[3]Laboratory of Organic Chemistry, Department of Chemistry and Applied Bioscience, ETH Zurich, 8093 Zurich, Switzerland

Correspondence to: Imad El Haddad (imad.el-haddad@psi.ch), Dongyu Wang (dongyu.wang@psi.ch), Houssni Lamkaddam (houssni.lamkaddam@psi.ch), Jay G. Slowik (jay.slowik@psi.ch)

**Abstract.** To elucidate the sources and chemical reaction pathways of organic vapors and particulate matter in the ambient atmosphere, real-time detection of both gas and particle phase is needed. State-of-the-art techniques often suffer from thermal decomposition, ionization-induced fragmentation, high cut-off size of aerosols or low time resolution. In response to all these limitations, we developed a new technique that uses extractive electrospray ionization (EESI) for online gas and particle chemical speciation, namely the dual-phase extractive electrospray ionization time-of-flight mass spectrometer (Dual-Phase-EESI-TOF or Dual-EESI in short). The Dual-EESI was designed and optimized to measure gas- and particle-phase species with saturation vapor concentrations spanning more than 10 orders of magnitude with good linearity and a measurement cycle as fast as 3 min. The gas-phase selectivity of the Dual-EESI was compared with that of nitrate chemical ionization mass spectrometry. In addition, we performed organic aerosol uptake experiments to characterize the relative gas and particle response factors. In general, the Dual-EESI is more sensitive toward gas-phase analytes as compared to their particle-phase counterparts. The real-time measurement capability of the Dual-EESI for chemically speciated gas- and particle-phase measurements can provide new insights into aerosol sources or formation mechanisms, where gas-particle partitioning behavior can be determined after absolute parameterization of the gas/particle sensitivity.

## 1 Introduction

Atmospheric aerosols are small particles suspended in the air, also denoted particulate matter (PM). They cool the climate either directly by light scattering, or indirectly by acting as cloud condensation nuclei (CCN), influencing clouds' albedo and lifetime (Seinfeld and Pandis, 2016). Poor air quality associated with high levels of PM is a major public health problem, and is one of the five leading causes of premature deaths worldwide, along with high blood pressure, smoking, diabetes and obesity (Cohen et al., 2017). Human exposure to PM caused ~8.9 million deaths, or ~10% of the total global burden of mortality in 2015 (Burnett et al., 2018), which is more than car accidents, HIV and malaria combined. Aerosols from direct emissions are classified as primary, while those produced from various atmospheric oxidation reactions are classified as secondary. The secondary aerosol may either condense onto existing particles or nucleate to induce new particle formation (NPF). NPF and particle growth happen on time scales of minutes to hours and involve millions of different oxidation products (Seinfeld and Pandis, 2016). To elucidate the sources and chemical reaction pathways of these products as well as their gas-particle partitioning behavior, rapid online detection of both gas and particle phase with high chemical resolution is needed (Nozière et al., 2015).

Online chemical characterization in the gas and particle phase is traditionally realized by using different mass spectrometers in parallel (Hoffmann et al., 2011). Chemical ionization mass spectrometry (CIMS) and proton-transfer-reaction mass

spectrometry (PTR-MS) with relatively low ionization energy (< 10 eV) are applied for gas-phase measurements (Hansel et al., 1999; Munson et al., 1966), whereas electron impact ionization and laser ablation mass spectrometry with high energy (> 70 eV) are applied for particle-phase measurements (Jayne et al., 2000; Noble et al., 1996). Because chemical speciation, ionization selectivity and efficiency between the different analytical techniques do not overlap, our understanding of the

dynamics between the two phases (gas and particle) is limited with such an approach (Riva et al., 2019). Efforts to detect of both phases using a single mass spectrometer led to the development of the Micro-Orifice Volatilization Impactor (MOVI), the Filter Inlet for Gases and Aerosols (FIGAERO), and the Thermal Desorption Differential Mobility Analyzer (TD-DMA) which can be coupled to CIMS (Brüggemann et al., 2015; Lopez-Hilfiker et al., 2014; Voisin et al., 2003; Wagner et al., 2018; Yatavelli et al., 2012, 2014; Yatavelli and Thornton, 2010), PTR-MS (Holzinger et al., 2010), or a standalone PTR-MS

combined with a particle inlet system such as CHARON-PTR (Eichler et al., 2015). Most of these semi-continuous techniques have an atmospherically relevant detection limit (a few ng m$^{-3}$), but their performance in molecular speciation is complicated by ongoing reactions on the collection substrate (Pratt and Prather, 2012; Timkovsky et al., 2015), as well as thermal decomposition due to thermal treatment of the collected particulate matter or/and ionization-induced fragmentation (Stark et al., 2017; Müller et al., 2017).

In response to the limitations of state-of-the-art techniques, an extractive electrospray ionization time-of-flight instrument (EESI-TOF) was developed to chemically characterize particles with sizes as small as 10 nm at time resolutions > 5 Hz and a detection limit of a few ng m$^{-3}$, by exploiting the soft ionization nature of the electrospray (Lopez-Hilfiker et al., 2019; Surdu et al., 2021). This technique was further developed for online metal detection (Giannoukos et al., 2020), aerosol source apportionment (Qi et al., 2019; Stefenelli et al., 2019), and with recent development of coupling to an ultra-high resolution

mass spectrometry (Lee et al., 2020). Despite these efforts, gaseous chemical characterization has not been attempted using this technique even though it is technically possible (Zhao et al., 2017). To improve our understanding of physicochemical processes between the gas and particle phase, we developed an inlet for the EESI-TOF that allows for automated and alternating measurements of the gas and particle phase, namely a Dual-Phase-EESI-TOF (Dual-EESI). First, we show the proof-of-principle measurement and demonstrate measurement linearity of the Dual-EESI with chemical standards spanning almost 10

orders of magnitude in estimated saturation vapor concentration. Then we present results from flow tube measurements of oxidation products of biogenic and anthropogenic VOC precursors to validate the capability of the Dual-EESI that distinguishes gaseous and particulate analytes. The gas-phase measurements of the Dual-EESI were compared to well-established CIMS techniques for evaluation of linearity, selectivity and sensitivity. Finally, the gas-phase sensitivity response of the EESI was empirically parameterized for the first time.

**2 Experimental**

**2.1 Dual-Phase-EESI-TOF inlet description**

The Dual-EESI source couples the EESI ion source design of Lopez-Hilfiker et al. (2019) with a newly developed inlet which automatically cycles through three different sampling channels, denoted as total-phase channel (TP), particle-phase channel (PP), and particle filter channel (FP). As shown in Figure 1, an open channel (length $L$: 0.5 m; diameter $\emptyset$: 10 x 8 mm; surface

to volume $S/V_{tube}$ = 500 m$^{-1}$) samples both gas-phase and particle-phase analytes (referred to as the total-phase channel, TP). An open tube ($L$: 0.7 m; $\emptyset$: 10 x 8 mm, 2 90˚ bends) with two multichannel charcoal denuders which remove gas-phase analytes is used for particle-phase analyte sampling (referred to as the particle-phase channel, PP). The charcoal denuders ($L$ = 2 x 0.048 m, $\emptyset$ = 36 mm with approximately 200 channels with an area dimension of 1.5 x 1.5 mm, $S/V_{demuder}$ = 533333 m$^{-1}$, Ionicon GmbH, Austria) have a high gas removal efficiency as shown in Figure S1 of the Supplement (~99% at 3 L min$^{-1}$ and

~97% at 11 L min$^{-1}$). Lastly, a HEPA filter is added in the particle-phase channel prior to the two multichannel charcoal denuders to remove particles for particle-phase background measurement, referred to as the particle filter channel (FP). The

gas-phase and particle-phase signals are calculated as TP - PP and PP - FP, respectively. The sampling flow of the EESI-TOF capillary is 0.8 L min$^{-1}$ whereas the sampling flow of the Dual-Phase-EESI-inlet (prior to the EESI-TOF capillary) was varied from 2 - 11 L min$^{-1}$ with a residence time ranging from 0.2 - 1 s to maximize gas and particle sampling efficiency (Figure S2, Figure S3 and Figure S4) depending on the experimental setups. The residence time in the PP is 1.4 times longer than in the TP. These residence times among sampling channels are optimized to be as short as possible while maintaining symmetric sampling between channels, minimization of gaseous diffusion losses and particulate re-volatilization which typically has a time scale of 10$^2$ minutes (Vaden et al., 2011).

We tested four different tubing materials—conductive perfluoroalkoxy alkane (PFA), conductive polytetrafluoroethylene (PTFE), stainless steel 306 and coated stainless steel 306—to determine desorption responses of semi-volatile compounds (SVOCs) in the Dual-EESI inlet as shown in Figure S5. Unless specified otherwise, stainless steel 306 tubing coated with functionalized hydrogenated amorphous silicon (SilcoNert 2000, Silcotek GmbH, Germany) was chosen for its rigidness, which allows for stable particle transmission and a lower adsorption rate in comparison to uncoated stainless steel 306 (Pagonis et al., 2017). A pressure sensor is added upstream of the electrospray region because pressure fluctuations > 5 mbar destabilize the electrospray (ES) signal. For high sampling flow applications (> 5 L min$^{-1}$, $\emptyset = 40$ mm, $L = 0.5$ m), it is recommended to use multiple particle filters in parallel to minimize flow impedance (i.e. pressure drop) as characterized in Figure S6. In this study, a single HEPA filter was used and the excess sampling flow rate was maintained at 5 L min$^{-1}$ unless specified.

Throughout the whole experiment, acetonitrile/H$_2$O (50/50 v/v) doped with 100 ppm NaI was used as the ES working solution. A potential difference of around 2.6-2.9 kV relative to the MS interface was applied to the ES solution, and an air pressure difference of 300 to 600 mbar was applied to the ES solution bottle reservoir, delivering 2.7 – 5.4 µl min$^{-1}$ of ES solution via a polyimide fused silica capillary (o.d.: 369 µm and i.d.: 75 µm, BGB Analytik, Switzerland). The ES droplets intersected with the sample analytes before entering the heated TOF-capillary kept at 275 ℃ (<1 ms residence time), undergoing a Coulomb explosion as the ES droplets evaporated. The ions generated from organic molecules were detected predominantly (> 95 % relative abundance) as sodiated adducts ([M+Na]$^+$) in the positive ionization mode by the TOF. The raw mass spectra (1 Hz) were post-averaged every 10 seconds using Tofware (version 2.5.13). All measured analyte signals were normalized by the most abundant electrospray ion (i.e. [NaI+Na]$^+$) to account for the variation of the electrospray signal ($\pm$ 5 %).

## 2.2 Ambient aerosol and gaseous surrogates

A permeation source containing a 5-ml glass vial filled with 0.1 g of solid camphor was used to generate camphor vapors. A flow rate of 1.5 L min$^{-1}$ permeation source vapor was mixed with 1.5 L min$^{-1}$ nebulized levoglucosan particles and introduced into the Dual-EESI for proof-of-principle measurements as shown in Figure S7. Calibrations of particles were carried out individually for organic (sebacic acid, levoglucosan and sucrose) and inorganic (iodic acid and ammonium nitrate) compounds. The flow from a nebulizer (1.5 L min$^{-1}$) was mixed with a dilution flow of zero air at 3, 5, 7, and 9 L min$^{-1}$ in a laminar mixing dilution unit. The mixture inside the laminar mixing dilution unit was core-sampled by the Dual-EESI. The laminar mixing dilution unit was operated with a slight overpressure to prevent re-entrainment of ambient analytes as shown in Figure S8. The particle mass concentration was measured by a scanning mobility particle sizer (SMPS, TSI Inc., USA) which comprises an electrostatic classifier 3082 with a differential mobility analyzer 3081 (TSI Inc., USA) and a condensation particle counter 3775 (TSI Inc., USA).

Gas standards (ethyl acetate, ethyl propionate, 3-pentanone, camphor, hydroxyacetone) for the Dual-EESI calibration were generated by a Vaporjet (Microfab Technologies Inc., USA, Figure S9) and measured by a proton-transfer-reaction time-of-flight mass spectrometer (PTR-TOF, Ionicon GmbH, Austria). In the Vaporjet, a single droplet of a dissolved chemical was periodically (20 - 200 Hz) dispensed by a piezoelectric microjet with an inner diameter of 50 µm. The droplets are continuously and fully flash-evaporated by a thermally regulated resistive heater at a temperature around 250 – 275 ºC. The number of

droplets per second is monitored by a stroboscopic camera with a magnifying lens (Verkouteren et al., 2006). All chemical solutions for the Vaporjet dispenser were prepared at a concentration of 0.1 M using Milli-Q water (18 MΩ cm).

Finally, we investigated the measurement linearity of gas-phase species in the presence of particle-phase species by mixing an increasing camphor gas concentration with a constant flow of particles (Figure S10). Solutions of 0.5 ppm of levoglucosan and iodic acid were prepared in Milli-Q water and mixed. This mixture solution was nebulized at 1.5 L min$^{-1}$ and mixed with 1.2 L min$^{-1}$ of camphor in a laminar mixing dilution unit before entering the Dual-EESI inlet. The concentration of camphor in the laminar mixing dilution unit was decreased by increasing the carrier flow (3-10 L min$^{-1}$) through the camphor permeation
source.

## 2.3 Biogenic and anthropogenic SOA

Measurements of atmospherically relevant gas and particulate oxidation products were performed in two types of experiments. In the first experiment (Figure S11), a 104-cm long Pyrex glass tube of 3.7 cm ID with a normalized residence time of 4.47 min$^2$ L$^{-1}$ was used as an oxidation flow reactor for monoterpene ozonolysis as described in Molteni et al. (2018). α-Pinene
vapor was generated via evaporation of the liquid compounds from a 1-L bubbler immersed in a 25 °C water bath, and carried by a stream of 2-10 mL min$^{-1}$ synthetic air (80:20 N$_2$:O$_2$) to provide a constant source of α-pinene (2 - 18 ppmv) inside the flow tube, as monitored by a quadrupole proton transfer reactor mass spectrometer (Q-PTR, Ionicon GmbH, Austria). To generate ozone, dry synthetic air (1 L min$^{-1}$) was irradiated by an amalgam lamp (185 & 254 nm; WISAG GmbH, Switzerland) and diluted by a factor of 10 to produce ∼280 ppbv of ozone, as measured by a Serinus 10 ozone analyzer (Ecotech Group,
Australia). The ozone flow (1 L min$^{-1}$) was combined at an angle of 90° with the α-pinene vapor (1 L min$^{-1}$) before entering the flow reactor. 9 L min$^{-1}$ zero air at 40% RH and 20 °C was introduced into the flow tube giving a total residence time of approximately 28 s for SOA formation. The Dual-EESI was used to measure the gaseous and particulate chemical composition. An SMPS was used to quantify the SOA mass concentration.

In the second experiment (Figure S12), a flow of 1 L min$^{-1}$ 1,3,5-trimethylbenzene vapor from a VOC permeation source was
mixed with 4 L min$^{-1}$ dry and 4 L min$^{-1}$ wet air. This gas mixture was combined with 1 L min$^{-1}$ of ozone at a mixing ratio of approximately 10 ppmv. After the introduction of this mixture into the first flow tube with a residence time of approximately 22 s, OH radicals were generated from ozone by irradiation with the UV-B lamps surrounding the flow tube, producing gaseous oxidation products of 1,3,5-trimethylbenzene. These gaseous oxidation products were continuously injected in the second flow tube concentrically for about 2 hours until the measured oxidation products were at steady-state, i.e. conditioning of the flow
tube wall was complete. After that, increasing concentrations of ammonium sulfate particles were injected into the center of the second flow tube by an Apex E desolvating nebulizer (Elemental Scientific, US) connected to a SC-2X autosampler (Elemental Scientific, US) which drew from 5 ammonium sulfate solutions with increasing concentrations as sources for seed particles (see Table S1), to provide the condensation sink for low-volatility oxidation products at an RH of 50-60 % and temperature of 23 ± 0.1 °C.

The gaseous chemical species were characterized by a Nitrate-CIMS and a PTR-TOF. The Nitrate-CIMS is composed of two main parts: a CI Eisele type-inlet for chemical ionization at ambient pressure (Eisele and Tanner, 1993) and an atmospheric pressure interface (API) time-of-flight mass spectrometer (Bertram et al., 2011; Ehn et al., 2010; Jokinen et al., 2012). Here, the CI-inlet was used with nitric acid where the nitric acid vapor was photoionized by a soft x-ray source (Hamamatsu L9490). As a result, the gaseous chemical species were detected after deprotonation, or cluster formation with the reagent ions such as
NO$_3^-$. The sampling flow of Nitrate-CIMS was diluted by a factor of 50 to minimize primary ion (e.g. NO$_3^-$) depletion. The PTR-TOF-MS used in this work is described in detail elsewhere (Graus et al., 2010). The PTR-TOF-MS was run with H$_3$O$^+$ as the reagent ion and used with the following parameters: 0.2 L min$^{-1}$ sample flow rate through a heated (333 K) capillary PEEK tubing (ID = 1.5 mm), 10 s time resolution, drift tube pressure = 2.2 mbar, drift tube temperature = 333 K, drift tube voltage = 550 V, resulting in an electric field (E/N) value of 120 Td (1 Td = 10$^{-17}$ cm$^2$ V$^{-1}$ ). H$_3$O$^+$ reagent ions donate a proton

to the gaseous chemical species within the air sample stream (via exothermic proton transfer reactions) producing mostly $[M+H]^+$ quasi-molecular compound ions before separation in the TOF (Breitenlechner et al., 2017).

## 3.0 Results and discussion

Figure 1 shows the sampling setup for the Dual-EESI inlet with three different sampling channels: (1) TP (blue channel), (2) PP (green channel) and (3) FP (green and red channels). All sampling pathways lead to the same electrospray ionization (ESI) chamber, where the gaseous and/or particulate analytes intersect with charged droplets discharged by an ESI capillary for subsequent ionizations. In the EESI scheme, when particles intersect with the charged ES droplets, they can be extracted into the aqueous phase and subsequently ionized via Coulomb fissions of these ES charged droplets. The gas molecules can be ionized by gas-phase ions formed during charged ES droplet evaporation, which is termed secondary electrospray ionization (SESI) (Wu et al., 2000). This approach was used in the design of Zhao et al. (2017), where charged ions were generated first before mixing with neutral analyte gasses inside a cross-flow ion-molecule reaction tube. According to the settings of our Dual-EESI inlet, which does not use a heated sheath flow to air ES droplet evaporation, analytes intersect with the ES droplets at ambient pressure. Therefore, EESI (i.e. analyte-droplet interaction) is expected to dominate in TP and PP channels, though SESI (i.e. ion-vapor interaction) may still occur in all channels including FP where gas desorption from the tubing wall coexists.

## 3.1 Proof-of-principle measurement

Figure 2 shows a typical measurement cycle of the Dual-EESI for two different sampling cycles of 1 and 0.1 background-to-sampling measurement period ratios. Each complete sampling cycle proceeds as follows: $FP_0$-TP-$FP_1$-PP-$FP_2$, where $FP_0$ is the $FP_2$ measurement from the previous measurement cycle. Both camphor and levoglucosan with concentrations of approximately 500 pptv and 10 $\mu g\ m^{-3}$ respectively, were measured during TP measurements (blue shade). During the combined $FP_1$-PP-$FP_2$ period in Figure 2a, the camphor signal decreased by a factor of 50. Lastly, during the FP measurement (red shade), the levoglucosan intensity decreased by at least a factor of 5. As shown in Figure 2a, the background level ($FP_0$) of the camphor signal increased from 1% of TP (before any camphor injection) to 1.5 % after injection. Figure 2b shows that when the duration of the PP measurement phase (during which camphor gas was denuded) was increased to 10 minute, the final background camphor signal intensity (~1.5 %) was similar to that observed with only 1 minute of PP measurement (Figure 2a). To check the performance and memory effect of the background measurement, we applied different background-to-sampling measurement period ratios for the sampling cycles as shown in Figure S13 and Table S2. The background levels (FP) of camphor (TP) and levoglucosan (PP) varied within the range of 1.2-1.5 % and 11-22 %, respectively for all these sampling sequences. The increase in the background level of levoglucosan can be attributed to the potential gas desorption of levoglucosan from the tubing walls downstream of the denuders (Lee et al., 2020; Lopez-Hilfiker et al., 2019). The background signals of the Dual-EESI vary by less than a factor of 2 for all the tested sampling sequences and are reproducible during $FP_1$ and $FP_2$ measurements. Similarly, for semivolatile compounds generated from precursor oxidation (Figure 4 and Figure S18), the average signal measured during $FP_1$ and $FP_2$ periods are lower than 15% and 10% of the TP signals, respectively. Averaged $FP_2$ measurements remain consistent for different sampling durations (Figure S13), indicating that background buildup is negligible. The rise time from $FP_0$ (i.e. $FP_2$ of the previous cycle) to TP as shown in Figure 2 is very short (< 1 min) compared to that observed using ion-molecule reaction (IMR) CIMS (Palm et al., 2019), where it was required to carry out fast zero measurements to account for vapor-wall interactions. For the Dual-EESI inlet, such effects are likely minor given the fast rise time and consistency of $FP_2$. Therefore, we use $FP_2$ as our background measurement. This is similar to the background treatment of single-phase EESI (Lopez-Hilfiker et al., 2019), which follows a $FP_0$-PP-$FP_1$ sampling scheme and uses the average of $FP_0$ and $FP_1$ as the background, where $FP_0$ is the $FP_1$ measurement of the previous cycle. As a result, the Dual-EESI

is able to provide quasi-simultaneous gas- and particle-phase measurements for standard compounds with measurement cycle periods as low as 5 minutes, including background correction. This capability was further tested below using environmentally relevant complex mixtures of gases and particles.

## 3.2 Influence of transmission for the quantification

Sampling flow conditions have been shown to affect the measurements of TP and PP via vapor-wall interactions and size-dependent particle transmission (Brown et al., 2021; Deming et al., 2019; Lee et al., 2021; Pagonis et al., 2017). In this section, we characterize the effect of response of gaseous and particulate compounds in TP and PP channels.

First, we investigate the influence of the vapor-wall interaction in the TP channel on the Dual-EESI gas sampling efficiency ($\sim$ 40 % at 1 L min$^{-1}$, see Figure S2 and Figure S3a). Different sampling flows of camphor were applied to the TP channel. Figure S3b shows that the background (FP measurement) relative to the TP measurement remains similar when the sampling flow of camphor decreases from 9 L min$^{-1}$ (higher sampled gas concentration) to 3 L min$^{-1}$ (lower sampled gas concentration). The similar camphor background level relative to TP at different gas sampling efficiencies could be attributed to the positioning of the valves downstream of the Dual-EESI TP and PP channels, where the remaining tubing surface area is approximately a factor of 7 less than the whole inlet, enabling minimal vapor-wall interactions inside the inlet.

Gas-wall interactions are especially important for semi-volatile and intermediate volatility organic compounds (Pagonis et al., 2017; Deming et al., 2019; Liu et al., 2019). We tested four different tubing materials—conductive perfluoroalkoxy alkane (PFA), conductive polytetrafluoroethylene (PTFE), stainless steel 306 and coated stainless steel 306 for desorption responses semi-volatile compounds in the Dual-EESI inlet. Compounds were generated from the OH oxidation of 1,2,4-trimethylbenzene at 30 % RH and 20 ˚C. The oxidation product signals were measured by an Eisele-type atmospheric pressure chemical ionization inlet (Nitrate-CIMS) connected after the Dual-EESI inlet at 10 L min$^{-1}$ sampling flow. Figure S5 shows that, under our sampling configurations, semi-volatile vapors ($C_9H_{12}O_{4-5}$) quickly equilibrates regardless of the tubing materials. This is most likely because the wall losses are dominated by the presence of other surfaces. The rapid equilibration times of ~3 minutes are likely due to the high flow used which ensures a combination of shorter residence time and higher dilution compared to previous studies (Pagonis et al., 2017; Deming et al., 2019; Liu et al., 2019), where gas-wall interactions were more severe and dependent on wall material. A similar behavior is observed for semi-volatile oxidation products of α-pinene + ozone ($C_{10}H_{16}O_3$ and $C_{10}H_{16}O_5$, Figure 4), measured by the Dual-EESI at 5 L min$^{-1}$ sampling flow.

To compare the particle signal between TP and PP, we investigated the change of detected particulate levoglucosan signal for the TP and PP channels at 7 and 9 L min$^{-1}$ after characterizing the particle transmission efficiencies of different channels (Figure S4). We found that the measured levoglucosan signal in PP is approximately 13-17 % lower than in TP, corresponding to a decrease of the transmitted particle mass concentration between TP and PP of 4-20 % as measured by an SMPS. This difference of the detected EESI signal between PP and TP has to be taken into account for quantifying the particle- and gas-phase concentrations. For instance, during sampling of smaller particles that exhibit higher sensitivity (Lee et al., 2021), we expect larger differences between PP and TP, and thus larger uncertainties in the quantification of gas-phase concentrations.

## 3.3 Measurement linearity

Several studies have reported that the ionization efficiency of chemical compounds in the gas and particle phase can differ due to different ionization mechanisms (Kumbhani et al., 2018; Law et al., 2010; Lee et al., 2021; Wang et al., 2012; Wingen and Finlayson-Pitts, 2019). One study suggested that the EESI sensitivity depends on volatility (Meier et al., 2011). To evaluate this, we used two groups of chemical compounds classified by volatility to investigate the effect observed by Meier et al. (2011). Group 1 consists of compounds with estimated saturation vapor concentrations $> 10^2$ µg m$^{-3}$, i.e., ethyl acetate, ethyl propionate, 3-pentanone, camphor and hydroxyacetone. Group 2 consists of compounds with estimated saturation vapor concentrations $< 10^1$ µg m$^{-3}$, i.e., sebacic acid, levoglucosan, iodic acid, ammonium nitrate and sucrose. These compounds

have been observed in ambient air and span almost 10 orders of magnitude in estimated saturation vapor concentration. Figure 3a and Figure 3b show that the Dual-EESI provides a linear response for gases and particles with a sensitivity range of approximately 2–3 orders of magnitude, consistent with other reports (Lopez-Hilfiker et al., 2019; Kruve et al., 2013). The detected ions in the gas or particle phase are measured as the same adduct ions, enabling the subtraction of TP-PP signal for gas-phase measurement. Organic compounds are mostly (> 95 %) detected by the EESI as $Na^+$-adducts. Unlike for inorganic salts and organic molecules with highly polar anions (e.g. sebacic acid) in either the gas or particle phase, the corresponding cation or carboxyl is partially or fully substituted by $Na^+$ before being associated with another $Na^+$ such as $(Na^+ X^-)Na^+$ or $(RCOO^- Na^+)Na^+$. For the investigated chemical standards, we do not observe an effect of such additional cation substitution on the response linearity. However, the possible effect of the adduct formation after cation substitution process on the measurement sensitivity and linearity should be investigated further in the future.

Figure 3 shows that the Dual-EESI exhibits a linear response to the individual gaseous and particulate analytes. However, this linearity could potentially be affected if the gaseous and particulate analyte mixtures interact with each other rather than just with the ES ions, leading to matrix effects such as ion enhancement or suppression via different mechanisms (Rovelli et al., 2020). Therefore, we characterized the linearity of a mixture of gaseous (camphor) and particulate (levoglucosan and iodic acid) compounds. Figure S15 shows that the relative signals of levoglucosan and iodic acid particles were negligibly affected (± 10 %) when the concentration of camphor gas was decreased by a factor of 10. The negligible matrix effect of gas and particle mixture measurement using the Dual-EESI is a promising feature for separating gaseous and particulate chemical species. We further assess the linearity of the gas-phase signals in Section 3.6 (Figure 6) for more complex mixtures, by comparing the EESI signals to those measured with Nitrate-CIMS.

## 3.4 Separation of gas and particle measurements using SOA

While experiments, discussed above, using chemical standards show that the Dual-EESI can measure gas and particle phases separately, they do not reflect the complexity of compounds present in the real atmosphere in the particle and gas phases. For this reason, we measured biogenic and anthropogenic oxidation products generated in the flow tube using two different sets of experiments. In the first set of experiments, SOA was produced from α-pinene (AP) ozonolysis with increasing production rate of the oxidation products, leading to an increasing SOA formation rate and condensation sink. As shown in Figure 4a, four stages of SOA mass concentration were generated by reacting ozone (< 280 ppbv) with increasing AP concentrations (2-18 ppmv) in excess. The condensation sink (CS) increased from 0.002 to 0.11 $s^{-1}$ (Figure S16) due to the increase in the AP SOA formation rate with SOA mass concentrations of 6 - 247 μg $m^{-3}$ (Figure S17). Figure 4b shows that the Dual-EESI gas-phase measurements of $C_{10}H_{14}O$ and $C_{10}H_{16}O$ behave commensurately with the measurements of the Q-PTR. To check for condensation of semi-volatile species, the time series of $C_{10}H_{16}O_3$ and $C_{10}H_{16}O_5$ are shown in Figure 4c-d. While the particle-phase signal of $C_{10}H_{16}O_5$ (green shade) increases with the SOA mass concentration, the particle-phase signal of $C_{10}H_{16}O_3$ is indiscernible from the background within our uncertainties and in the studied range of CS.

In the second set of experiments, we measured the oxidation products of 1,2,4-trimethylbenzene (TMB) through OH oxidation, where the gas-phase production rate was kept constant, while the condensation sink was increased by varying the mass concentration of ammonium sulfate (Table S1). As shown in Figure S18, the TP signal of $C_9H_{12}O_5$ decreases with increasing CS, while producing an increase in the PP signal at higher CS. Among the TMB oxidation products, $C_9H_{10-12}O_4$ are the most abundant species in both phases. Figure S19 shows that the relative abundance of more oxygenated compounds (O = 5-6) in the gas phase is lower than in the particle phase, similar to the case of α-pinene (Figure S17). Our comparison of 73 chemical species (Table S3) from AP and TMB SOA in Figure S20 demonstrates the ability of the Dual-EESI that separates gas- and particle-phase components almost simultaneously, i.e. can resolve particle-phase species increase with increasing gas-species production and CS in the first set of experiments, and particle-phase species increase due to increasing gas-species condensation with an increasing condensation sink in the second set of experiments.

## 3.5 Biogenic and anthropogenic SOA constituents

Figures 3 and 4 show that the Dual-EESI can separate gas-phase and particle-phase components from chemical standards as well as from SOA mixtures. To demonstrate the different behavior of species in the gas- and particle-phase, Figure 5 shows their relative abundance in the gas-phase (TP-PP) and particle-phase (PP-FP) as a function of oxygen and carbon number. The $C_{8-10}H_{12-16}O_{1-8}$ and $C_{7-9}H_{8-16}O_{2-8}$ species were chosen for the AP and TMB system, respectively, because these species had already been reported for these systems (Tsiligiannis et al., 2019; Wang et al., 2021; Zhang et al., 2015). The total intensity of the gas-phase measurement is at least 20 times higher than the total intensity of the particle-phase measurement for both AP and TMB systems. However, a 20 times higher gas-phase intensity for a detected species does not necessarily mean that its gas-phase concentration is 20 times higher than the particle-phase concentration. A large fraction of this difference may be attributed to differences in the Brownian coagulation coefficient of gas molecules and particles with ES droplets, which for particles yields a significant increase in sensitivity with decreasing particle size for particles with diameter $D_p$ <100 nm (Lee et al., 2021). To directly compare gas-phase to particle-phase measurements in terms of absolute concentration, molecular calibration and correction for gas/particle sensitivity response are still required (see Section 3.7) (Kruve et al., 2013; Mayhew et al., 2020; Wang et al., 2021). Since the intensities in Figure 5 are not corrected, we only compare the measured relative abundances (i.e. not absolute concentrations) of those oxygenated species between the gas-phase and particle-phase measurements.

The left panel in Figure 5 compares $C_{8-10}$ species of AP SOA. The gas-phase species mostly contain 2-4 O atoms, whereas the particle-phase species mostly contain 5-8 O atoms with an SOA mass concentration of 45 µg m$^{-3}$. Such relative abundance distribution is also consistent for different AP SOA mass concentrations (6, 45, 123, 246 µg m$^{-3}$) as shown in Figure S17 where the relative abundance of different species in the gas- and particle-phase vary commensurately with the SOA mass concentration. For the $C_{7-9}$ species of TMB SOA at 31.4 µg m$^{-3}$ (right panel of Figure 5), the relative abundances of $C_{7-9}H_{10-12}O_{5-6}$ species in the particle phase are higher than in the gas phase. $C_9H_{12-16}O_4$ are the most abundant species in both phases, indicating their semi-volatile nature. Consistently, when decreasing the CS from 0.16 to 0.08 s$^{-1}$, the total intensity of the gas-phase measurement increases and the total intensity of the particle-phase measurement decreases as depicted in Figure S19. The particle-phase measurement in AP is much more oxygenated than its gas phase when compared to TMB gas- and particle-phase measurement. This difference could be primarily attributed to the dynamic where AP SOA CS is generated due to the increase in the oxidation product concentration whereas it is constant for TMB SOA. Overall, the Dual-EESI measures species over a wide volatility range in both the gas and particle phase, and giving insights into the different partitioning behaviors of compounds spanning a wide range of volatilities with increasing molecular weight or/and functionalization (Donahue et al., 2011; Pankow and Asher, 2008).

## 3.6 Linearity and sensitivity comparison to the CIMS

As demonstrated in Figure 3, the Dual-EESI shows good measurement linearity for chemical standards measured in both the gas and particle phases. To potentially elucidate the gas-to-particle partitioning processes, measurement linearity and sensitivity of semi-volatile species with saturation vapor concentration within 0.1-100 µg m$^{-3}$ is crucial (Donahue et al., 2012). Thus, a series of semi-volatile organic compounds generated via TMB OH oxidation ($C_9H_{12,14,16}O_{3-8}$) were compared between the Dual-EESI and Nitrate-CIMS, as shown in Figure 6. Generally, the Dual-EESI detects oxygenated products ranging from $O_1$ to $O_8$ but is insensitive to non-oxygenated VOCs. Figure 6a shows that Na$^+$-adduct formation in the Dual-EESI in the gas phase is linear with respect to the two different ionization pathways of the Nitrate-CIMS, where some analytes undergo two different ionization pathways (Riva et al., 2019). The Dual-EESI measures a wider range than either the deprotonated ions ($O_3$-$O_5$) or the nitrate adduct ions ($O_3$-$O_7$) measured by Nitrate-CIMS. Furthermore, Figure 6b shows that the relative response factor of the Dual-EESI vs CIMS spans over roughly two orders of magnitude, with Nitrate-CIMS being more sensitive towards more oxygenated compounds, which has been reported in previous studies (Lamkaddam et al., 2021; Riva et al.,

2019). Wang et al. (2021) reported that the ionization efficiency of EESI increases by two orders of magnitude as the volatility decreases by eight orders of magnitude indicating that the Dual-EESI exhibits a shallower sensitivity dependence on the oxygen number than Nitrate-CIMS. A systematic discussion of gas- and particle-phase sensitivity for the Dual-EESI is given in Section 3.7.

**3.7 Dual-EESI gas- and particle-phase sensitivity response**

Given a steady gas-phase production rate (i.e. constant precursor injection rates), the decrease in total-phase signal in response to an increasing condensation sink (Figure S21), which facilitates condensation of organic vapors, suggests that the Dual-EESI has higher sensitivity towards gas-phase species than towards particle-phase species. Taking the sampling efficiencies (> 60 %, Figure S4) and size-dependent sensitivity (~20 % decrease for a $D_p$ increase from 90 to 130 nm, Figure S22, Eqs. S2-S3, Table S1) into account, Figure 7 shows that the increase in the particle-phase signals after seed particle injection for the same molecule is on average 11 times lower than the decrease in the gas-phase signals, i.e, the relative gas- and particle-phase sensitivity response factor differ by up to a factor of 11, with the gas-phase being more sensitive. This is reproducible for 5 different condensation sink conditions (0.075–0.16 s$^{-1}$) where 75 % of the mean of (change of gas)/(change of particle) signals are 3 times higher than standard deviation as shown in Figure S23. The calculation of the particle- and gas-phase signals shown in Figure 7 is detailed in Section S6 of the Supplement. The spread in Figure 7 indicates that the saturation vapor concentration alone is insufficient to describe the differences in gas vs. particle relative response factors for the Dual-EESI. Other factors such as diffusivity may also contribute, the unclear correlation ($R^2 = 0.115$) between the gas/particle sensitivity ratio and binary diffusivity values estimated using only molecular formulae (Fuller et al., 1996) could be caused by the lacking knowledge of analyte structures. This higher sensitivity in the gas phase compared to the particle phase is consistent with the strong size-dependence of the EESI response (Figure S22 and Lee et al., 2021) and most likely results from the increase in the coagulation rate coefficients between the ES droplets and the analyte particles or gas molecules as their size decreases. The higher coagulation rate from the gaseous molecule in EESI results in a higher detection (extraction) limit of gas-phase measurement in comparison to particle-phase measurement.

**4 Conclusion**

We developed and characterized a novel instrument, the Dual-Phase-EESI-TOF (Dual-EESI) which can measure gaseous and particulate chemical species quasi-simultaneously with the same ionization and detection system. The inlet design, materials, and sampling sequence are optimized to achieve accurate background-correction and high sampling efficiency. Negligible measurement interferences between gas and particle were demonstrated for both constant and varying gas and particle concentrations using chemical standards and SOA mixtures. In addition, the Dual-EESI can measure a wide range of oxygenated species for $1 < O < 8$ in the gas phase, allowing for a more comprehensive characterization of ambient gas and aerosols. Using the Na$^+$-adduct formation for gas- and particle-phase species measurements, the Dual-EESI simplifies the gas and particle-phase composition analysis, which could provide a self-consistent method to deduce SVOC volatility. The measurement cycle can be as fast as 5 min, with 1 Hz time resolution, which is useful for atmospheric chemistry and aerosol dynamics investigations. We also present the relative sensitivity of gaseous versus particulate species where the quantity of the gas-phase can be estimated using the particle-phase calibration within a factor of 11 uncertainty. This factor of 11 likewise dominates the uncertainty in estimating the partitioning coefficient for SVOC (0.3 μg m$^{-3}$ ≤ $c^*$ ≤ 300 μg m$^{-3}$ under typical ambient conditions (C$_{OA}$~10 μg m$^{-3}$). Future work should focus on absolute response factor parameterization between the gaseous and particulate species, and a systematic elucidation of their sensitivities can help to determine partitioning coefficients in real-time, shedding the light to source apportionment of total organics and partitioning investigation for specific sources and/or pollution.


*Data availability.* Data presented in this study can be obtained at the Zenodo online repository hosted by CERN (https://doi.org/10.5281/zenodo.6331419, Lee et al., 2021). Raw data can be obtained by contacting the corresponding authors.

*Author contributions.* C.P.L., J.S., J.D., G.W., P.A.S. designed the inlet. C.P.L., G.W., P.A.S., M.S., X.Z., D.W. fabricated the
inlet. C.P.L., I.E., J.D., H.L., M.S., D.W., D.B. designed the experiment. M.S., C.P.L., D.W., H.L, J.D., X.Z., A.B., S.G. performed the experiments. C.P.L., M.S., H.L., D.W. analyzed the data. C.P.L., I.E., M.S., D.W., H. L., J. D., J.S., D.B., U.B., A. S. H. P., X. M. interpreted the compiled results. C.P.L. prepared the manuscript. All authors contributed to the discussion and revision of the manuscript.

*Competing interests.* The authors declare that they have no conflict of interest.

*Acknowledgements.* We would like to thank Rene Richter for his dedicated commitments during the development of this work and PSI workshop in providing the infrastructure for parts fabrication. We would also like to thank Liwei Wang and Yandong Tong for their advice on PTR and AMS data analysis, respectively. We also appreciate the supervision of Julia Schmale which
helped in the implementation of this development. Lastly, thanks to Felipe Lopez-Hilfiker and Veronika Pospíšilová for the discussions during the initial stage of this development.

*Financial support.* This research has been supported by Horizon 2020 (grant nos. PSI-FELLOW-II-3i (grant no. 701647) and EUROCHAMP-2020 (grant no. 730997)) and the Schweizerischer Nationalfonds zur Förderung der Wissenschaftlichen
Forschung (grant nos. 200020_172602 and BSSGI0_155846).

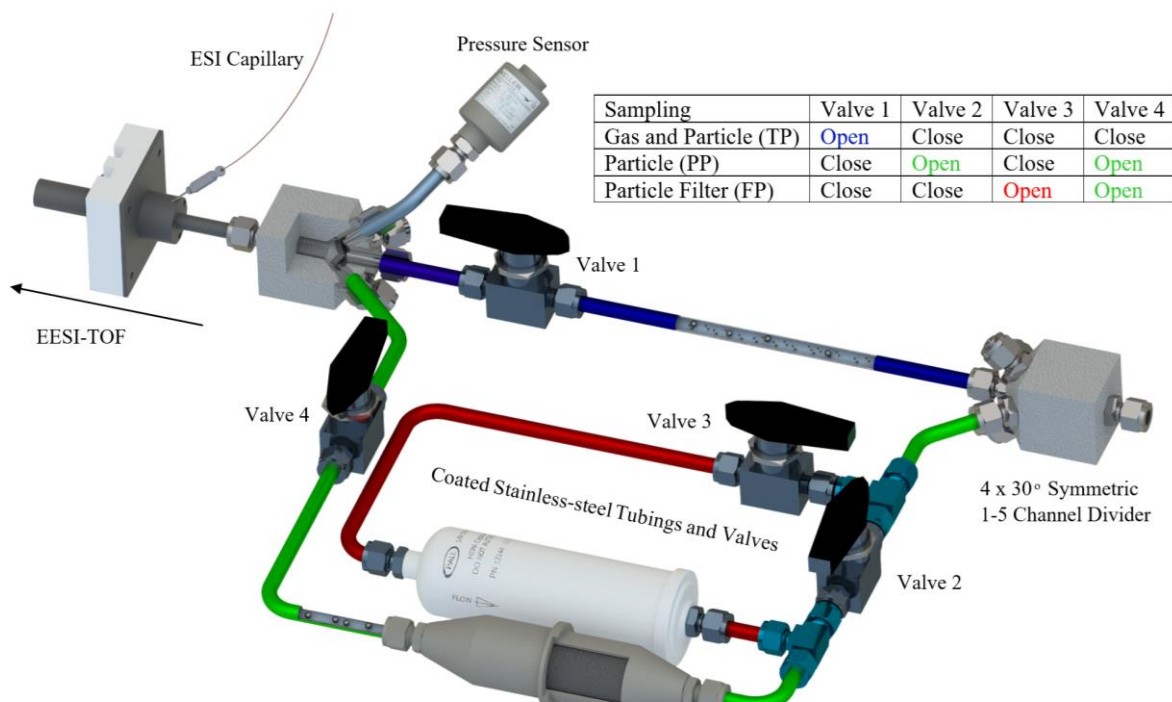

| Sampling | Valve 1 | Valve 2 | Valve 3 | Valve 4 |
|---|---|---|---|---|
| Gas and Particle (TP) | Open | Close | Close | Close |
| Particle (PP) | Close | Open | Close | Open |
| Particle Filter (FP) | Close | Close | Open | Open |

**Figure 1. Schematic diagram of the developed ionization inlet for the Dual-EESI. The inlet consists of the following channels: total-phase (TP, shaded blue); particle-phase (PP, shaded green); and particle filter (FP, shaded green and red). The gas-phase measurement is obtained as TP - PP, whereas the particle-phase measurement is obtained as PP – FP$_2$.**


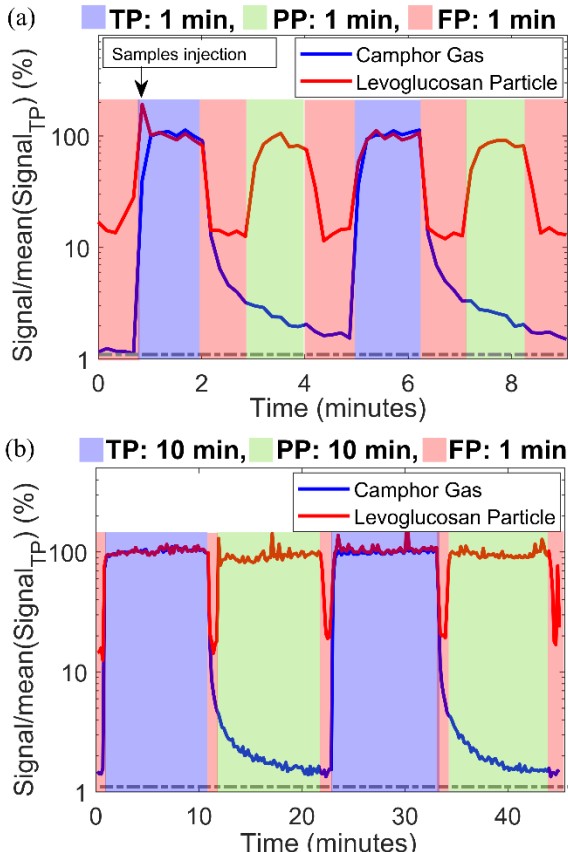

**Figure 2. Dual-EESI measurements of gaseous camphor and particulate levoglucosan for two different ratios of the background-to-sampling measurement period (a) 1 and (b) 0.1. Please note that the camphor in (b) was injected at the first minute in (a), thus both sampling conditions have the same level of background (1 %). The relative changes of the signals in percentage were obtained after**

**normalizing the signals by the averaged TP signal. Dashed grey lines denote the initial background level before the injection of camphor and levoglucosan. Different sampling time ratios (background to measurement time) are shown in Figure S13 and Table S2. Data were collected at a time resolution of 1 s, and then averaged to 10 s for display.**

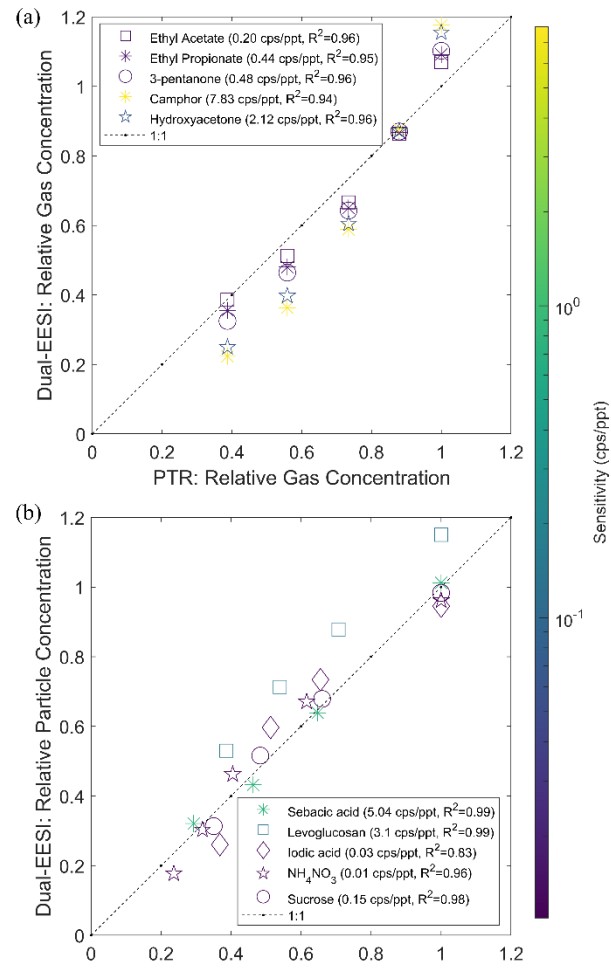

**Figure 3. Dual-EESI linearity for chemical standards that span almost 10 orders of magnitudes in the estimated saturation vapor concentration. (a) Measured intensity of different gaseous species (TP-PP) and (b) particulate species (PP-FP$_2$) with estimated saturation concentration from $10^5$ to $10^{-15}$ µg m$^{-3}$. The gaseous species were generated by a Vaporjet and quantified by a PTR (see Figure S9 and Figure S14). The particulate species were generated by a nebulizer and quantified by an SMPS using the effective density of the chemical standard. Color scales indicate the sensitivity (cps ppt$^{-1}$) in the gas and particle phase as indicated in the respective legend. Each set of calibration data points is fitted by a linear regression $p_1 \cdot x$. For display, the y-axis data points are then normalized by the maximum of the regressed data points using the x-axis data points.**

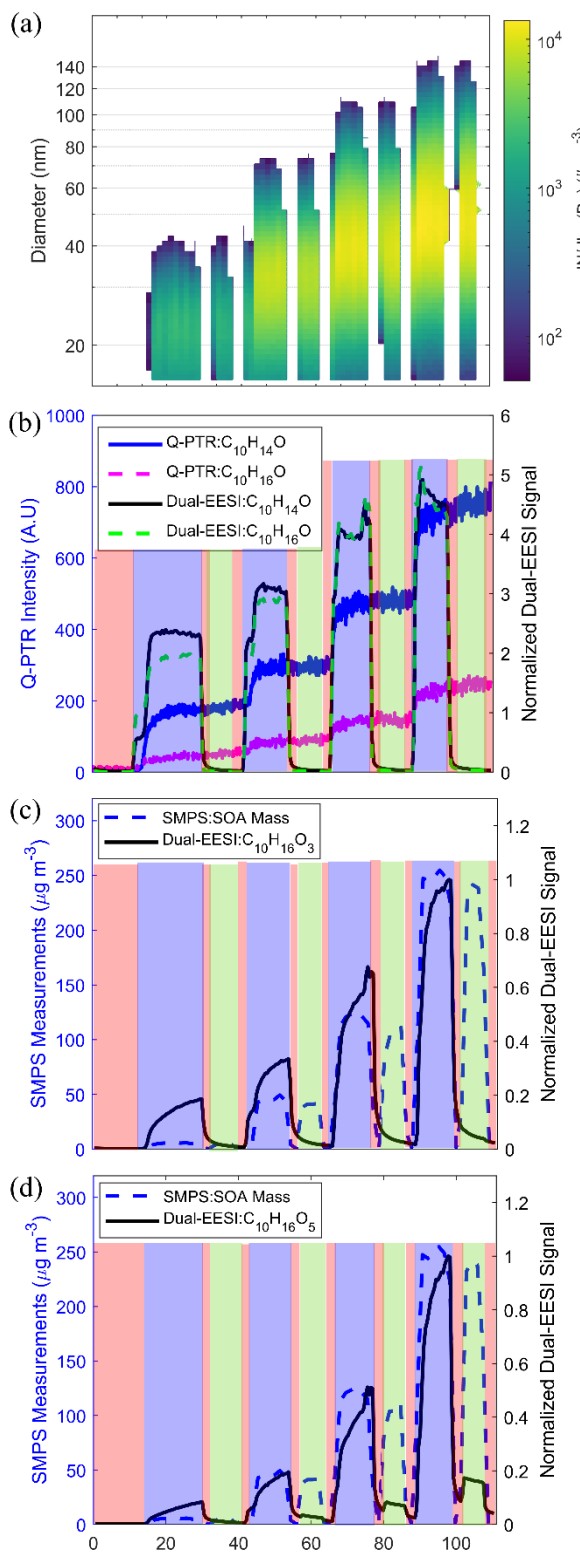

**Figure 4. (a)** SMPS size distribution for four experimental stages, each with different SOA mass concentrations (regions where $dN/dlog(D_p)$ < 50 $cm^{-3}$ are replaced by blank space). **(b)** Comparison of the Dual-EESI and Q-PTR measurements for $C_{10}H_{14,16}O$. Normalized Dual-EESI signal of **(c)** $C_{10}H_{16}O_3$ and **(d)** $C_{10}H_{16}O_5$ for increasing SOA mass concentration. Each sampling cycle consists of 10 - 15 minutes of TP (blue), 5 minutes of $FP_1$ (red), 10 minutes of PP (green) and 5 minutes of $FP_2$ (red) measurements. Note that the switching of the sampling channel for FP and PP only started when the signals of $C_{10}H_{16}O_{3-5}$ reached a steady state in the TP. Thus, the sampling time of the first TP phase of each cycle varied for these four stages.

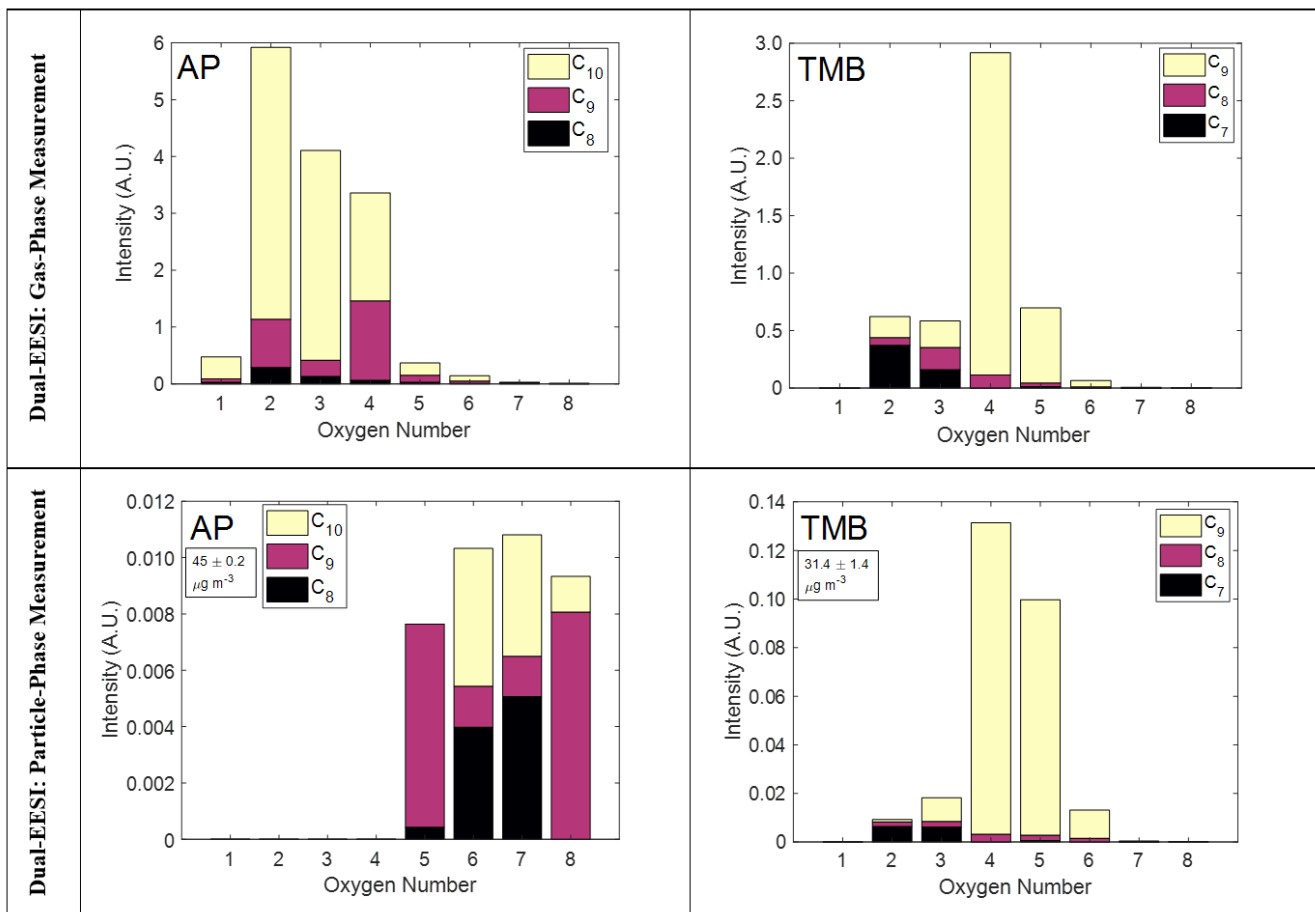

**Figure 5. Measurement of gas- and particle-phase oxidation product mixtures generated inside a flow tube reactor using the Dual-EESI. (Left panel) Gas-phase oxidation products (top) and SOA (bottom) from AP ozonolysis for $C_{8-10}H_{12-16}O_{1-8}$ with a total number of 73 species. (Right panel) Gas-phase oxidation products (top) and SOA from TMB oxidation with OH for $C_{7-9}H_{8-16}O_{2-8}$ with a total number of 32 species (see Table S3). Corrections for particle transmission efficiencies were not applied here, which might cause an uncertainty up to 15 %, as discussed in Section 3.2. The A.U. denotes the quantity of the ion intensity after normalization to the most abundant ES ion $[NaI+Na]^+$ which was applied to the gas- and particle-phase measurements.**



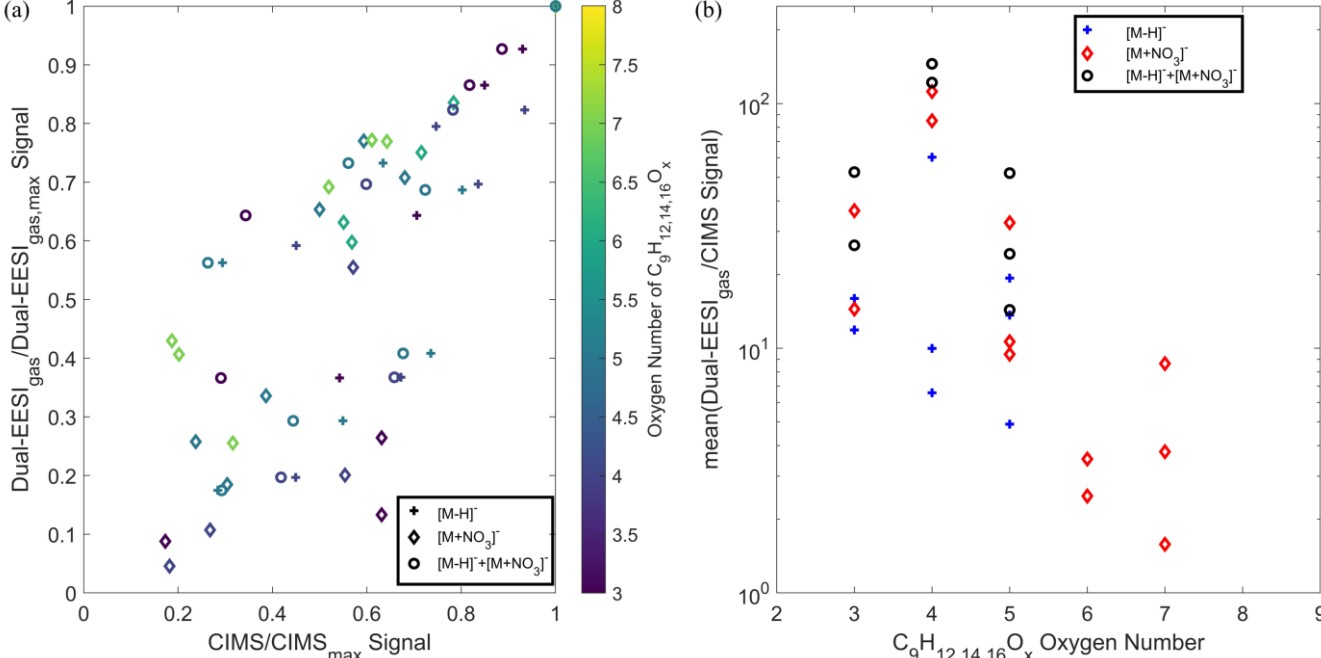

**Figure 6. Comparison between the Dual-EESI and Nitrate-CIMS for a series of TMB oxidation products in the gas phase. Since some analytes detected in Nitrate-CIMS might have two ionization pathways via deprotonation or nitration, for each analyte M, we compared [M-H]⁻, [M+NO₃]⁻ and [M-H]⁻+ [M+NO₃]⁻ from Nitrate-CIMS to the [M+Na]⁺ of the Dual-EESI. Please note that summation of detected ions in CIMS was only implemented for ions which can be detected in both [M-H]⁻ and [M+NO₃]⁻ of the same compound, where their respective sensitivities in Nitrate-CIMS may differ. (a) Normalized signal of the Dual-EESI and the Nitrate-CIMS to their respective maxima where the color scale indicates the oxygen number. (b) Averaged intensity ratio between the Dual-EESI in the gas phase and Nitrate-CIMS for TMB products with different oxygen numbers (C₉H₁₂,₁₄,₁₆O₃₋₈).**


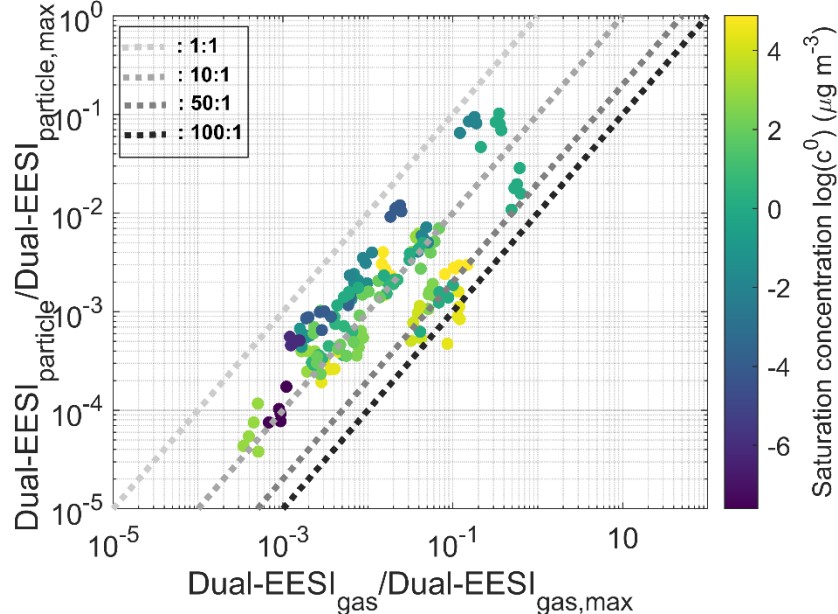


**Figure 7. Relative gas-to-particle sensitivity response of the Dual-EESI.** The change (decrease) in gas-phase signal and change (increase) in particle-phase signal from different molecules (Table S4) are shown on a logarithmic scale due to the 2-3 orders of magnitudes RIE variability range among the molecules for each phase. 1:1, 10:1, 50:1, 100:1 dotted lines are added for visualization of the gas-to-particle phase sensitivity response. The particle-phase measurements were corrected for size-dependent sensitivity as

shown in Figure S22. The data points are color scaled by their estimated saturation vapor concentration (Donahue et al., 2011).

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
