# Peer review of "High-Frequency Gaseous and Particulate Chemical Characterization using Extractive Electrospray Ionization Mass Spectrometry (Dual-Phase-EESI-TOF)"

_Atmospheric Measurement Techniques, 2021_

## Author Comment (AC1)

We thank the reviewers for their comments. The original comments are in normal font type. Our point-by-point responses are shown below in ***bold italic***. Manuscript changes are shown in blue.

RC1 (*https://doi.org/10.5194/amt-2021-325-RC1*)

Summary:

The authors introduce a new inlet called the Dual-EESI that combines with a TOF mass spectrometer to alternately measure both gas and particle phases over the span of several minutes. Several flow tube experiments were performed and compared with CIMS measurements to show the proof of concept. They find that the EESI is more sensitive to gas phase compounds than particle phase. Future work will be to develop quantification methods for both phases. Generally, I believe this inlet will be a powerful addition to the EESI technique. But to make that true, I believe the background signal (specifically the signal due to desorption from tubing and EESI walls) needs to be better considered and characterized for sticky or semivolatile compounds (which are a focus of the manuscript). Failure to do so could lead to substantial measurement biases for such compounds. I think this will require major revisions, and perhaps an adjustment to the inlet configuration or sampling method. Once the backgrounds are properly considered, the Dual-EESI will certainly be a valuable inlet and would be appropriate for publication in AMT.

***We thank the reviewer for his comments and respond below to his concerns about the performance of the inlet for semi-volatile compounds.***

Main comments:

Line 92 and Fig. S5: This figure is used to draw the conclusion that there is essentially no difference in gas-wall interactions across the four tubing types. However, I do not think this experiment was a great way of testing that, because it did not isolate the tubing response. Instead, the measurements are combing the responses resulting from the UV lamps warming up (in the two left hand panels), the residence time distribution of the two flow tubes (especially smearing the decay in the right hand panels), and any wall interactions in the inlet/IMR of the acetate CIMS being used to sample the compounds. Please address the following related to this comment:

-The wall interactions in the acetate CIMS could be negligible if sampling with the Eisele-type IMR at atmospheric pressure, but could be considerable if sampling in a low pressure IMR; please specify.

***We used an Eisele-type atmospheric pressure chemical ionization inlet with nitric acid reagent ion in this study as mentioned at line 161. This is now also mentioned in the caption of Figure S5. The new figure caption reads as follows:***

"The oxidation product signal was measured by an Eisele-type atmospheric pressure chemical ionization inlet (Nitrate-CIMS) connected after the Dual-EESI inlet at 10 L min$^{-1}$ sampling flow.*"*

-I think what Fig. S5 (especially when turning UV off) does suggest is that these two particular compounds are reaching equilibrium with the wall surfaces of all tubing types on a time scale that is less than the 2-3 min decay time needed to flush out the flow tubes. So while you can't say whether there are differences between tubing types, this essentially shows that the tubing type does not matter for these two compounds when sampling this flow tube setup because the dominant wall effects are elsewhere (and obscured by the lamp warming up or the flow tubes flushing out). My guess is that the tubing wall effects are not dominant because 1) the residence time in that tubing is short, minimizing the adsorption or absorption to the wall relative the compounds that get sampled without first interacting with a wall, and 2) any compounds that desorb from the wall get diluted in the fast (2-10 liter per minute) flow, so concentrations remain relatively low. For comparison, the flow rates used in e.g. Deming et al 2019 and Pagonis et al 2017 were much slower (often 0.3 liters per min) through a meter or more of tubing, leading

to much larger wall effects. I suggest you discuss these details in your manuscript in the paragraph at line 92, otherwise please explain if you have a different interpretation. Also, Palm et al. 2019 (https://doi.org/10.5194/amt-12-5829-2019) is another resource that may be helpful for thinking about tubing/inlet wall effects along with the Deming, Pagonis, and Liu papers.

*We agree with the reviewer's assessments and interpretations. We used high flows, similar to CIMS, to minimize the residence time in the inlet and thus the adsorption of compounds onto the inlet walls. We moved the part of Figure S5 discussion in experimental section 2.1 to the result section 3.2.  The revised text reads as follows:*

*In Section 2.1, line 89-97:*

"We tested four different tubing materials—conductive perfluoroalkoxy alkane (PFA), conductive polytetrafluoroethylene (PTFE), stainless steel 306 and coated stainless steel 306—to determine desorption responses of semi-volatile compounds (SVOCs) in the Dual-EESI inlet as shown in Figure S5. Unless specified otherwise, stainless steel 306 tubing coated with functionalized hydrogenated amorphous silicon (SilcoNert 2000, Silcotek GmbH, Germany) was chosen for its rigidness, which allows for stable particle transmission and a lower adsorption rate in comparison to uncoated stainless steel 306 (Pagonis et al., 2017). A pressure sensor is added upstream of the electrospray region because pressure fluctuations $> 5$ mbar destabilize the electrospray (ES) signal.  For high sampling flow applications ($> 5$ L min$^{-1}$, $\varnothing = 40$ mm, $L = 0.5$ m), it is recommended to use multiple particle filters in parallel to minimize flow impedance (i.e. pressure drop) as characterized in Figure S6. In this study, a single HEPA filter was used and the excess sampling flow rate was maintained at 5 L min$^{-1}$ unless specified."

*In Section 3.2, line 219-230:*

"Gas-wall interactions are especially important for semi-volatile and intermediate volatility organic compounds (Pagonis et al., 2017; Deming et al., 2019; Liu et al., 2019). We tested four different tubing materials—conductive perfluoroalkoxy alkane (PFA), conductive polytetrafluoroethylene (PTFE), stainless steel 306 and coated stainless steel 306 for desorption responses semi-volatile compounds in the Dual-EESI inlet. Compounds were generated from the OH oxidation of 1,2,4-trimethylbenzene at 30 % RH and 20 ˚C. The oxidation product signals were measured by an Eisele-type atmospheric pressure chemical ionization inlet (Nitrate-CIMS) connected after the Dual-EESI inlet at 10 L min$^{-1}$ sampling flow. Figure S5 shows that, under our sampling configurations, semi-volatile vapors ($C_9H_{12}O_{4-5}$) quickly equilibrates regardless of the tubing materials. This is most likely because the wall losses are dominated by the presence of other surfaces. The rapid equilibration times of ~3 minutes are likely due to the high flow used which ensures a combination of shorter residence time and higher dilution compared to previous studies (Pagonis et al., 2017; Deming et al., 2019; Liu et al., 2019), where gas-wall interactions were more severe and dependent on wall material. A similar behavior is observed for semi-volatile oxidation products of $\alpha$-pinene + ozone ($C_{10}H_{16}O_3$ and $C_{10}H_{16}O_5$, Figure 4), measured by the Dual-EESI at 5 L min$^{-1}$ sampling flow."

-Lastly, this analysis applies only to the two compounds shown in Fig. S5. It's not clear whether those compounds are particularly 'sticky' or not. Could there be 'stickier' compounds that would interact more with the walls, and then show important differences as a function of tubing type? I'm not asking for a full analysis across a range of C* values, but perhaps you could just state what the estimated C* values are for the two compounds you show? That could help put your measurements in context with the broader measurements in the Pagonis, Deming, Liu, Palm papers.

*We focused on semivolatiles measurement here because compounds that are stickier than $C_9H_{12}O_{4-5}$ might exist mostly in particle phase, leaving the gas-phase concentration below detection limit under the current experimental conditions. As requested, we added the estimated $\log(c^*)$ values for these two compounds in the caption of Figure S5 as "$\log(c^*) = 0.06$ and $-1.97$ µg m$^{-3}$, respectively, calculated from molecular formula according to (Donahue et al, 2011)".*

*We also noted in the main text result section 3.2 at line 229-230:*

*"A similar behavior is observed for semi-volatile oxidation products of α-pinene + ozone ($C_{10}H_{16}O_3$ and $C_{10}H_{16}O_5$, Figure 4), measured by the Dual-EESI at 5 L min$^{-1}$ sampling flow."*

Line 186: There are a few issues with this discussion of the camphor background percentage here. First, the 1.2-1.5% camphor signal measured at the end of the combined FP-PP-FP cycle is not the appropriate gas phase background to subtract from the amount of camphor sampled at the end of each TP mode. Again, I will suggest that you consider the framework presented in Palm et al. 2019 (https://doi.org/10.5194/amt-12-5829-2019). The total signal at the end of TP is the sum of the signal from compounds that are sampled directly without wall interaction plus the amount of background signal from compounds that had interacted with a surface (which depends on how much and how long a compound was sampled) and then desorbed to be sampled. That background signal is dynamic and depends on the sampling history. A more appropriate gas phase background to subtract would be that measured immediately after switching from TP to FP (after the residence time of the tubing has cleared). The concentration measured at that time will be a close approximation of the amount of background signal generated at the end of TP. For camphor in Fig. 2, it looks like the background signal could be ~10% or more of the total signal. For stickier compounds, this value could be even higher, for example it looks like ~30% of the signal for C9H12O8 in Fig. S18. By waiting to the end of the FP-PP-FP cycle, the background signal decays due to desorption from tubing and EESI wall surfaces while sampling clean (scrubbed) air, and is no longer relevant to the end of the prior TP. However, even the background measured shortly after switching from TP to FP is not completely accurate, because you are simultaneously changing the sources of background by switching the tubing the air is sampled through. During TP, the background can come from both the blue tubing and the EESI inner surfaces (including tubing between the blue tubing and the EESI). But when sampling FP, you cut out the background coming from the blue tubing and only sample that from the EESI (assuming the background from the walls downstream of the carbon denuder is negligible). Now, if the background coming from the blue tubing is negligible compared to the background coming from inside the EESI, then this doesn't matter and your background measurements will be close enough. But if the blue tubing is a substantial source of signal, this could be a major problem for sampling some sticky or semivolatile gases through this inlet. This could be tested by sampling a sticky compound and then injecting clean air for several seconds either upstream or downstream of the blue tubing to see if there is a difference (which would be due to background signal in the blue tubing). To summarize this lengthy comment, please carefully consider (and be explicit about) your background determination and subtraction methods, determine if the current inlet configuration and sampling method is sufficient, and make adjustments if necessary to reduce/remove these biases.

*We agree that measurement of the background is non-trivial. The two main considerations are the differences between sampling pathways when switching from TP to FP and desorption kinetics. Regarding the first issue, our analysis of the tubing material effect above suggests that the sampling tube-vapor interaction is likely minor, and that the background is dominated by surface-vapor interactions at the mass spectrometer inlet. The use of charcoal denuder in the green channel in Figure 1 (PP and FP) should not affect particle-related backgrounds, and also eliminates the effects of upstream sampling differences between pathways (e.g., desorption from HEPA filter). Therefore, the FP and PP channels should be applicable for TP background measurements. The appropriate background-correction strategy is an area of active debate in the literature, with studies such as Palm et al. (2019) preferring a fast zero-method (i.e. $FP_1$). We note that Palm et al., (2019) used an improved, coaxial ion-molecule reaction ionization region, which reduced vapor-wall interactions and shortened the signal rise time compared to more conventional IMR designs. In the present system, our sampling scheme follows $FP_0$-TP-$FP_1$-PP-$FP_2$, where $FP_0$ is the $FP_2$ from the previous sample*

*cycle. We show that the rise time from $FP_0$ to TP is very short compared to what has been reported for IMR-CIMS, indicating that vapor-wall interaction is minor. The background rapidly decays during $FP_1$ and remains relatively stable during $FP_2$. We also show that there is no appreciable change in $FP_2$ values (and where $FP_0 \approx FP_2$), when the sampling channels are being alternated during measurements. For these reasons, we prefer to use $FP_2$ as our background. We clarify our sampling schemes, the difference between two FP periods ($FP_1$ and $FP_2$), and our choice of background in the revised text.*

*In Section 3.1, line 182-184:*

"Each complete sampling cycle proceeds as follows: $FP_0$-TP-$FP_1$-PP-$FP_2$, where $FP_0$ is the $FP_2$ measurement from the previous measurement cycle. Both camphor and levoglucosan with concentrations of approximately 500 pptv and 10 μg m$^{-3}$ respectively, were measured during TP measurements (blue shade). During the combined $FP_1$-PP-$FP_2$ period in Figure 2a, the camphor signal decreased by a factor of 50."

*In Section 3.1, Line 194-207:*

"The background signals of the Dual-EESI vary by less than a factor of 2 for all the tested sampling sequences and are reproducible during $FP_1$ and $FP_2$ measurements. Similarly, for semivolatile compounds generated from precursor oxidation (Figure 4 and Figure S18), the average signal measured during $FP_1$ and $FP_2$ periods are lower than 15% and 10% of the TP signals, respectively. Averaged $FP_2$ measurements remain consistent for different sampling durations (Figure S13), indicating that background buildup is negligible. The rise time from $FP_0$ (i.e. $FP_2$ of the previous cycle) to TP as shown in Figure 2 is very short (< 1 min) compared to that observed using ion-molecule reaction (IMR) CIMS (Palm et al., 2019), where it was required to carry out fast zero measurements to account for vapor-wall interactions. For the Dual-EESI inlet, such effects are likely minor given the fast rise time and consistency of $FP_2$. Therefore, we use $FP_2$ as our background measurement. This is similar to the background treatment of single-phase EESI (Lopez-Hilfiker et al., 2019), which follows a $FP_0$-PP-$FP_1$ sampling scheme and uses the average of $FP_0$ and $FP_1$ as the background, where $FP_0$ is the $FP_1$ measurement of the previous cycle. As a result, the Dual-EESI is able to provide quasi-simultaneous gas- and particle-phase measurements for standard compounds with measurement cycle periods as low as 5 minutes, including background correction. This capability was further tested below using environmentally relevant complex mixtures of gases and particles."

Line 335: I believe this Dual-EESI inlet will be very valuable for being able to sample gases and particles at the same time. It's not clear to me, though, that it will be as useful for deducing SVOC volatilities. I believe that to be an extremely difficult task for any instrument or combination of instruments, just given measurement uncertainties for both gas and particle measurements. This may be too much to ask for this paper, but I'm wondering of you could provide an estimate of how large the error bars will be on a calculation of the C* value for a given compound? E.g. you could make assumptions about the relative sensitivities and measurement uncertainties for a compound in the gas vs. particle phase, and propagate the measurement uncertainties through to estimate the uncertainty on partitioning fraction or C*. My guess is the uncertainties in C* will be an order of magnitude or so, and will only be able to be estimated for the 2-3 or so order of magnitude range where semivolatile compounds are measurable in both phases above detection limit simultaneously. In the end, maybe it is enough to be able to say whether or not a compound is semivolatile. Any thoughts you have about the utility of the Dual-EESI in this respect would be useful to add to the manuscript, though again this is not to take away from the other very useful aspects of the Dual-EESI.

*We agree with the reviewer's assessment and have incorporated into our conclusion.*

*Line 360-361:* "Using the $Na^+$-adduct formation for gas- and particle-phase species measurements, the Dual-EESI simplifies the gas and particle-phase composition analysis."

*Line 364-366:* "We also compare the relative sensitivity of gaseous versus particulate species, which as shown in Fig. 7 are linearly related but with considerable spread. This introduces approximately a factor of 11 uncertainty in using the particle-phase calibration to estimate gas-phase concentrations. This factor of 11 likewise dominates the uncertainty in estimating the partitioning coefficient for SVOC ($0.3 \, \mu g \, m^{-3} \leq c^* \leq 300 \, \mu g \, m^{-3}$ under typical ambient conditions ($C_{OA} \sim 10 \, \mu g \, m^{-3}$)."

Line 346: The correlation of $R^2 = 0.115$ in Fig. S25 is quite low, so it does not seem to justify your conclusion here that higher gas relative to particle phase detection is correlated with diffusivity in air. Perhaps the estimation of diffusivity based on molecular formula (without knowledge of structure) leads to uncertainties that are too high in the diffusivity estimation? Anyway, I would suggest it would be fine to leave Fig. S25 out.

*We agree that the low $R^2$ value could be due to the high uncertainty of parameterization used to estimated molecular diffusivity without structural information. We have revised our discussion of Figure S25 accordingly and we have revised our discussion of Figure S25 as follows.*

*Line 344-348:* "The spread in Figure 7 indicates that the saturation vapor concentration alone is insufficient to describe the differences in gas vs. particle relative response factors for the Dual-EESI. Other factors such as diffusivity may also contribute, the unclear correlation ($R^2 = 0.115$) between the gas/particle sensitivity ratio and binary diffusivity values estimated using only molecular formulae (Fuller et al., 1996) could be caused by the lacking knowledge of analyte structures."

Technical comments:

Fig. 1: The legend is difficult to understand. I would suggest labelling the valves and identifying which ones are open or closed for each sampling method, rather than which sampling lines are open or closed. For instance, one of the valves on the green line has to be closed during particle filter mode. Also, particle filter mode is labeled as background in the legend, please make them the same to be consistent.

*We updated the Fig. 1 as requested for the labelling of the valves and background as shown below.*

[Figure]

Fig. S6: This figure didn't copy correctly.

*We re-rendered Figure S6.*

Line 168: Here also, be consistent with the labelling of each mode, i.e., use TP, PP, and FP instead of gas a particle, particle, and background measurement.

*We changed the labelling of each mode by using TP, PP and FP instead of gas and particle, particle and background measurement.*

Line 171: Isn't EESI not expected to happen in the FP mode, when the particles have been filtered out prior to sampling?

*We clarified the ionization setting at line 171-179: "*In the EESI scheme, when particles intersect with the charged ES droplets, they can be extracted into the aqueous phase and subsequently ionized via Coulomb fissions of these ES charged droplets. The gas molecules can be ionized by gas-phase ions formed during charged ES droplet evaporation, which is termed secondary electrospray ionization (SESI) (Wu et al., 2000). This approach was used in the design of Zhao et al. (2017), where charged ions were generated first before mixing with neutral analyte gasses inside a cross-flow ion-molecule reaction tube. According to the settings of our Dual-EESI inlet, which does not use a heated sheath flow to air ES droplet evaporation, analytes intersect with the ES droplets at ambient pressure. Therefore, EESI (i.e. analyte-droplet interaction) is expected to dominate in TP and PP channels, though SESI (i.e. ion-vapor interaction) may still occur in all channels including FP where gas desorption from the tubing wall coexists.*"

Line 308: missing ")" at the end of "(right panel of Figure 5".

*We added the missing ")".*

Fig. S12: The inlet should be labeled "Dual-Phase-EESI-Inlet" instead of ESI.

*We updated Fig. S12 using Dual-Phase-EESI-Inlet label as shown below.*

[Figure]

---

## Author Comment (AC2)

We thank the reviewers for their comments. The original comments are in normal font type. Our point-by-point responses are shown below in ***bold italic***. Manuscript changes are shown in blue.

RC2 (*https://doi.org/10.5194/amt-2021-325-RC2*)
* * *
This manuscript presents a novel mass spectrometric technique for studying gas- and particle-phase organic compounds with a single electrospray (ESI) ion source. The instrument operates by alternately sampling form three distinct channels: a direct inlet, an inlet where gas-phase compounds are denuded, and an inlet where particles are filtered, and gas-phase compounds are denuded. These three channels allow the authors to separate ESI background, from gas- and particle-phase signal by difference.

The technique works, is useful for studying the atmosphere, and is appropriate for AMT. My suggested revisions are below.

My main criticism is that this is a SESI + EESI instrument, and not a dual-EESI. These acronyms each correspond to distinct ionization pathways, and each have extensive literature describing the factors that affect their sensitivity. I think the authors will be best served by connecting their technique with the existing literature by calling this "dual ESI", "S/EESI", or anything that conveys that the method aims to separate EESI signal from SESI signal.

The novelty of this work is that the authors are bringing together two established techniques (SESI+EESI) in a single ion source. There are already SESI techniques for studying organic gases in the atmosphere that go beyond "technically possible", and the authors cite one example (Zhao et al 2017). That work already demonstrated linear SESI response for organics in the atmosphere, and even characterized the strong humidity dependence of SESI sensitivity.

Overall, I feel that there are significant details missing from this paper, covered in the additional comments below. The most critical area in need of discussion is the slow response time of the instrument to the gas-phase analytes, and how this response time affects the background subtraction. The authors largely avoid addressing this by sticking to raw time series instead of showing background-subtracted data, and since the background subtraction is central to the technique more detail is needed.

***In secondary electrospray ionization (SESI), ions generated by electrospray ionization are used to charge the analytes in the gas-phase. In the design used by Zhao et al. (2017), the electrospray is generated at ambient pressure, charged ions are generated after droplets pass through the evaporation region, and the ionization of neutral analytes occurs at reduced pressure in a cross-flow ion-molecule reactor. In our design, the gas and/or particle analytes intersect with an unheated electrospray at ambient pressure before entering the heated ion capillary of the mass spectrometer. Therefore, we expect that the droplet-vapor interaction is more dominant than gas-phase charge transfer, making the technique more EESI than SESI for gas-phase detection. The higher ionization efficiency of gases vs. particles follows the size dependence ionization efficiency of particles determined by Lee et al., 2021, where smaller particles (including gases) have higher ionization efficiency due to their higher collision efficiency with the charged droplets. We have added the following clarifications in the revised manuscript at line 171-179:***

In the EESI scheme, when particles intersect with the charged ES droplets, they can be extracted into the aqueous phase and subsequently ionized via Coulomb fissions of these ES charged droplets. The gas molecules can be ionized by gas-phase ions formed during charged ES droplet evaporation, which is termed secondary electrospray ionization (SESI) (Wu et al., 2000). This approach was used in the design of Zhao et al. (2017), where charged ions were generated first before mixing with neutral analyte gasses inside a cross-flow ion-molecule reaction tube. According to the settings of our Dual-EESI inlet, which does not use a heated sheath flow to air ES droplet evaporation, analytes intersect with the ES droplets at ambient pressure. Therefore, EESI (i.e. analyte-droplet interaction) is expected to dominate in TP and

PP channels, though SESI (i.e. ion-vapor interaction) may still occur in all channels including FP where gas desorption from the tubing wall coexists.

Line 70: What was the ESI working solution? Looks like AcN:Water and NaI dopant, and I'm guessing the solvent ratio is 1:1, but this needs to be added to methods

*We added the description of the ESI operational condition at line 98-106:*

"Throughout the whole experiment, acetonitrile/$H_2O$ (50/50 v/v) doped with 100 ppm NaI was used as the ES working solution. A potential difference of around 2.6-2.9 kV relative to the MS interface was applied to the ES solution, and an air pressure difference of 300 to 600 mbar was applied to the ES solution bottle reservoir, delivering $2.7 – 5.4$ µl $min^{-1}$ of ES solution via a polyimide fused silica capillary (o.d.: 369 µm and i.d.: 75 µm, BGB Analytik, Switzerland). The ES droplets intersected with the sample analytes before entering the heated TOF-capillary kept at 275 °C (<1 ms residence time), undergoing a Coulomb explosion as the ES droplets evaporated. The ions generated from organic molecules were detected predominantly (> 95 % relative abundance) as sodiated adducts ($[M+Na]^+$) in the positive ionization mode by the TOF. The raw mass spectra (1 Hz) were post-averaged every 10 seconds using Tofware (version 2.5.13). All measured analyte signals were normalized by the most abundant electrospray ion (i.e. $[NaI+Na]^+$) to account for the variation of the electrospray signal ($\pm$ 5 %)."

Line 166: $[M+H]^+$ ions for PTR-MS

*We changed "$M^{+1}$" to "$[M+H]^+$"as requested.*

Line 177: EESI only occurs if there are soluble particles present to be intercepted by the ESI drops. So in the FP channel, only ESI is occurring, since there are no gas- or particle phase analytes. In the PP channel signal is due to both ESI and EESI. In the TP channel signal is ESI + EESI + SESI

*ESI is usually meant as the infusion electrospray solution, this is not the case for our ES configuration. Thus, we clarified the ionization scheme at line 171-179:*

"In the EESI scheme, when particles intersect with the charged ES droplets, they can be extracted into the aqueous phase and subsequently ionized via Coulomb fissions of these ES charged droplets. The gas molecules can be ionized by gas-phase ions formed during charged ES droplet evaporation, which is termed secondary electrospray ionization (SESI) (Wu et al., 2000). This approach was used in the design of Zhao et al. (2017), where charged ions were generated first before mixing with neutral analyte gasses inside a cross-flow ion-molecule reaction tube. According to the settings of our Dual-EESI inlet, which does not use a heated sheath flow to air ES droplet evaporation, analytes intersect with the ES droplets at ambient pressure. Therefore, EESI (i.e. analyte-droplet interaction) is expected to dominate in TP and PP channels, though SESI (i.e. ion-vapor interaction) may still occur in all channels including FP where gas desorption from the tubing wall coexists."

Line 185: The FP measurements surrounding the TP measurement are very uneven, due to the slow response time of the camphor gas. The subtraction of FP from TP is critical for determining gas-phase signal later in the paper, and so the authors must discuss how they defined average FP signal for slow-responding gases.

**The average of the FP signal is defined as the average signal during the second FP measurement window, $FP_2$ in the TP-$FP_1$-PP-$FP_2$ cycle. We demonstrated in Figure S13 that the FP signal of camphor does not vary by more than few % within the background/sampling measurement period ratio of $0.1 – 1$ as discussed in line 190-193. Slow responding gases such as semivolatiles were addressed in Figures S5 and S18 where $C_9H_{12}O_{4-5}$ require 4 minutes to reach it previous background level. We have clarified our choice of background measurement in the revised text:**

*In Section 3.1, line 182-184:*

"Each complete sampling cycle proceeds as follows: $FP_0$-TP-$FP_1$-PP-$FP_2$, where $FP_0$ is the $FP_2$ measurement from the previous measurement cycle. Both camphor and levoglucosan with concentrations of approximately 500 pptv and 10 μg m$^{-3}$ respectively, were measured during TP measurements (blue shade). During the combined $FP_1$-PP-$FP_2$ period in Figure 2a, the camphor signal decreased by a factor of 50."

*In Section 3.1, Line 194-207:*

"The background signals of the Dual-EESI vary by less than a factor of 2 for all the tested sampling sequences and are reproducible during $FP_1$ and $FP_2$ measurements. Similarly, for semivolatile compounds generated from precursor oxidation (Figure 4 and Figure S18), the average signal measured during $FP_1$ and $FP_2$ periods are lower than 15% and 10% of the TP signals, respectively. Averaged $FP_2$ measurements remain consistent for different sampling durations (Figure S13), indicating that background buildup is negligible. The rise time from $FP_0$ (i.e. $FP_2$ of the previous cycle) to TP as shown in Figure 2 is very short (< 1 min) compared to that observed using ion-molecule reaction (IMR) CIMS (Palm et al., 2019), where it was required to carry out fast zero measurements to account for vapor-wall interactions. For the Dual-EESI inlet, such effects are likely minor given the fast rise time and consistency of $FP_2$. Therefore, we use $FP_2$ as our background measurement. This is similar to the background treatment of single-phase EESI (Lopez-Hilfiker et al., 2019), which follows a $FP_0$-PP-$FP_1$ sampling scheme and uses the average of $FP_0$ and $FP_1$ as the background, where $FP_0$ is the $FP_1$ measurement of the previous cycle. As a result, the Dual-EESI is able to provide quasi-simultaneous gas- and particle-phase measurements for standard compounds with measurement cycle periods as low as 5 minutes, including background correction. This capability was further tested below using environmentally relevant complex mixtures of gases and particles."

Fig 3: Should y-axis read "Relative Signal" instead of "Relative Gas Concentration"?

*Both figures are relative signal from EESI where the signals were generated by ionization of either gas-only or particle-only analytes. Thus, we labelled y-axis as "Relative Gas Concentration" and "Relative Particle Concentration" for clarity.*

Fig 3: There is clear and consistent structure to the residuals of gas-phase (SESI) sensitivity, and the authors must address this. I do not agree that this is a linear response for gas-phase analytes.

*While we agree with the reviewer that residuals seem to have a structure in Figure 3, the point does not deviate strongly from linearity with errors less than 20% (and $R^2$ = 0.94). In addition, we have little statistics to support a non-linear relationship. For example, the comparisons between the Dual-EESI and Nitrate-CIMS in Figure 6 indicate a linear relationship.*

Fig 6: I am unable to follow the significance of including the deprotonation ionization pathway from the nitrate CIMS. Are these data points all from a single experiment? Or did the authors run separate experiments with difference conditions in the Nitrate CIMS IMR? If it's all one experiment, then some acids are being detected as both [M+NO3]- and [M-H]- by the nitrate CIMS (e.g. C9H12O5, C9H14O5, and C9H16O5). Those two nitrate CIMS peaks would correspond to a single SESI peak [M+Na]+; so the authors should be summing the two nitrate CIMS signals in order to plot against the SESI signal, and not plotting the nitrate CIMS ionization mechanisms separately. There is no discussion in the text to help the reader understand the significance of the two ionization schemes, and the authors must add clarification.

*All data points were taken from a single experiment where the precursor concentrations were varied. The [M+NO$_3$]$^-$ and [M-H]$^-$ ions were observed in the same Nitrate-CIMS spectra. Following the reviewer's suggestion, we added the sum of the [M+NO$_3$]$^-$ and [M-H]$^-$ ion intensities in the figure when the same compound undergoes two ionization pathways. Since linearity is observed between*

*[M+NO₃]⁻ or [M-H]⁻ ions of Nitrate-CIMS to [M+Na]⁺ ions of the Dual-EESI, combined linearity is also expected for sum of the [M+NO₃]⁻ and [M-H]⁻ ions from CIMS as shown in the updated figure below.*

[Figure]

*We updated the caption that the two ionization pathways of Nitrate-CIMS for the same compound may not have the same sensitivities.*

*Updated caption for Figure 6*: "Comparison between the Dual-EESI and Nitrate-CIMS for a series of TMB oxidation products in the gas phase. Since some analytes detected in Nitrate-CIMS might have two ionization pathways via deprotonation or nitration, for each analyte M, we compared [M-H]⁻, [M+NO₃]⁻ and [M-H]⁻+ [M+NO₃]⁻ from Nitrate-CIMS to the [M+Na]⁺ of the Dual-EESI. Please note that summation of detected ions in CIMS was only implemented for ions which can be detected in both [M-H]⁻ and [M+NO₃]⁻ of the same compound, where their respective sensitivities in Nitrate-CIMS may differ. (a) Normalized signal of the Dual-EESI and the Nitrate-CIMS to their respective maxima where the color scale indicates the oxygen number. (b) Averaged intensity ratio between the Dual-EESI in the gas phase and Nitrate-CIMS for TMB products with different oxygen numbers ($C_9H_{12,14,16}O_{3-8}$)."

*Modified line 324-326:* "Figure 6a shows that Na⁺-adduct formation in the Dual-EESI in the gas phase is linear with respect to the two different ionization pathways of the Nitrate CIMS, where some analytes undergo two different ionization pathways (Riva et al., 2019)."

Line 335: There is no discussion of how the saturation concentration for Figure 7 is calculated. Is it based on the attributed molecular formula? What conclusion am I supposed to draw from the C* trend?

*The original figure caption cited Donahue et al. (2011) for the calculation of saturation concentration; we now clarify that this is based on molecular formulae. Using this calculation, we demonstrated that saturation concentration cannot explain the relative gas-to-particle sensitivity response of the Dual-EESI. This is now stated in the manuscript at line 344-348:*

"The spread in Figure 7 indicates that the saturation vapor concentration alone is insufficient to describe the differences in gas vs. particle relative response factors for the Dual-EESI. Other factors such as diffusivity may also contribute, the unclear correlation ($R^2 = 0.115$) between the gas/particle sensitivity ratio and binary diffusivity values estimated using only molecular formulae (Fuller et al., 1996) could be caused by the lacking knowledge of analyte structures."

Figure 7: are those actually arbitrary units? Or is that signal? The authors confidently discuss signal ratios, so I think this is signal, but that's not clear.

*We removed (A.U.) and updated Figure 7 label axis for clarity.*

[Figure]

Figure S6 did not render legibly in my pdf

*We re-rendered Figure S6.*